# General Proximal Incremental Aggregated Gradient Algorithms: Better and Novel Results under General Scheme[*]

**Tao Sun**
College of Computer
National University of Defense Technology
Changsha, Hunan 410073, China
`nudtsuntao@163.com`

**Yuejiao Sun**
Department of Mathematics
University of California, Los Angeles
Los Angeles, CA 90095, USA
`sunyj@math.ucla.edu`

**Dongsheng Li**[†]
College of Computer
National University of Defense Technology
Changsha, Hunan 410073, China
`dsli@nudt.edu.cn`

**Qing Liao**
Department of Computer Science & Technology
Harbin Institute of Technology (Shenzhen)
Shenzhen, Guangdong 518055, China
`liaoqing@hit.edu.cn`

## Abstract

The incremental aggregated gradient algorithm is popular in network optimization and machine learning research. However, the current convergence results require the objective function to be strongly convex. And the existing convergence rates are also limited to linear convergence. Due to the mathematical techniques, the stepsize in the algorithm is restricted by the strongly convex constant, which may make the stepsize be very small (the strongly convex constant may be small).

In this paper, we propose a general proximal incremental aggregated gradient algorithm, which contains various existing algorithms including the basic incremental aggregated gradient method. Better and new convergence results are proved even with the general scheme. The novel results presented in this paper, which have not appeared in previous literature, include: a general scheme, nonconvex analysis, the sublinear convergence rates of the function values, much larger stepsizes that guarantee the convergence, the convergence when noise exists, the line search strategy of the proximal incremental aggregated gradient algorithm and its convergence.

## 1 Introduction

Many problems in machine learning and network optimization can be formulated as

$$\min_x \{F(x) = f(x) + g(x)\}, \tag{1}$$

where $f(x) = \sum_{i=1}^m f_i(x)$, $x \in \mathbb{R}^n$, $f_i$ is differentiable, $\nabla f_i$ is Lipschitz continuous with $L_i$, for $i = 1, 2, \ldots, m$, and $g$ is proximable. A state-of-the-art method for this problem is the proximal gradient method [10], which requires to compute the full gradient of $f$ in each iteration. However, when the number of the component functions $f_i$ is very large, i.e. $m \gg 1$, it is costly to obtain the full

---

[*]This work is sponsored in part by the National Key R&D Program of China under Grant No. 2018YFB0204300 and the National Natural Science Foundation of China under Grants (61932001 and 61906200).

[†]Dongsheng Li is the corresponding author.

gradient $\nabla f$; on the other hand, in some network cases, calculating the full gradient is not allowed, either. Thus, *the incremental gradient algorithms* are developed to avoid computing the full gradient.

The main idea of the incremental gradient descent lies on computing the gradients of partial components of $f$ to refresh the full gradient. Precisely, in each iteration, it selects an index set $S$ from $\{1, 2, \ldots, m\}$; and then computes $\sum_{i \in S} \nabla f_i$ to update the full gradient. It requires much less computation than the gradient descent without losing too much accuracy of the true gradient. It is natural to consider two index selection strategies: deterministic and stochastic. In fact, all the incremental gradient algorithms for solving problem (1) can be labeled as one of these two routines.

## 1.1 The general PIAG algorithm

Let $x^i$ denote the $i$-th iterate. We first define a $\sigma$-algebra $\chi^k := \sigma(x^1, x^2, \ldots, x^k)$. Consider a general proximal incremental aggregated gradient algorithm which performs as

$$\begin{cases} \mathbb{E}(v^k \mid \chi^k) = \sum_{i=1}^m \nabla f_i(x^{k-\tau_{i,k}}) + e^k, \\ x^{k+1} = \mathbf{prox}_{\gamma_k g}[x^k - \gamma_k v^k], \end{cases} \tag{2}$$

where $\tau_{i,k}$ is the delay associated with $f_i$ and $e^k$ is the noise in the $k$-th iteration. The first equation in (2) indicates that $v^k$ is an approximation of the full gradient $\nabla f$ with delays and noises in the perspective of expectation. For simplicity, we call this algorithm as the general PIAG algorithm.

## 1.2 Literature review

As mentioned before, by the strategies of index selection, the literature can also be divided into two classes.

*On the deterministic road*: Bertsekas proposed the *Incremental Gradient* (IG) method for problem (1) when $g \equiv 0$ [4]. To obtain convergence, IG method requires diminishing stepsizes, even for smooth and strongly convex functions [5]. A special condition was proposed in [26] to relax this condition. The second order IG method was also developed in [13]. An improved version of the IG is the *Incremental Aggregated Gradient* (IAG) method [6, 29]. When $f_i$ is a quadratic function, the convergence based on the perturbation analysis of the eigenvalues of a periodic dynamic linear system was given by [6]. The global convergence is proved in [29]; and if the local Lipschitzian error condition and local strong convexity are satisfied, the local linear convergence can also be proved. [1] established lower complexity bounds for IAG for problem (1) with $g \equiv 0$. The linear convergence rates are proved under strong convexity assumption [12, 30].

*On the stochastic road*: The pioneer of this class is the *Stochastic Gradient Descent* (SGD) method [20], which suggests picking $i$ from $\{1, 2, \ldots, m\}$ in each iteration with uniform probability, and using $m \nabla f_i$ to replace $\nabla f$. However, SGD requires diminishing stepsizes, which makes its performance poor in both theory and practice. Due to the large deviation of $m \nabla f_i$ from $f$, "variance reduction" schemes were proposed later, such as SVRG method [14], SAG method [25], and the SAGA method [11] are developed. With selected constant stepsizes, linear convergence has been proved in the strongly convex case, and ergodic sublinear convergence has been proved in the non-strongly convex case.

## 1.3 Relations with existing algorithms

In this part, we will present several popular existing algorithms which can be covered by the general PIAG.

E.1. *(Inexact) Proximal Gradient Desdent Algorithm*: When $\tau_{i,k} \equiv 0$ and expectation vanishes, the general PIAG is equivalent to $x^{k+1} = \mathbf{prox}_{\gamma_k g}[x^k - \gamma_k \sum_{i=1}^m \nabla f_i(x^k) + e^k]$.

E.2. *(Inexact) Proximal Incremental Aggregated Gradient Algorithm*: In the $k$-th iteration, pick $i_k$ essentially cyclicly and then update $x^{k+1}$ as $x^{k+1} = \mathbf{prox}_{\gamma_k g}[x^k - \gamma_k(\nabla f_{i_k}(x^k) + \sum_{i \neq i_k} \nabla f_i(x^{k-\tau_{i,k}}) + e^k)]$, where

$$\begin{cases} \tau_{i,k+1} = \tau_{i,k} + 1 \text{ if } i \neq i_k, \\ \tau_{i,k+1} = 1 \text{ if } i = i_k. \end{cases} \tag{3}$$

In each iteration, one just needs to compute $\nabla f_{i_k}(x^k)$ and the term $\sum_{i \neq i_k} \nabla f_i(x^{k-\tau_{i,k}})$ is shared by the memory.

E.3. *Deterministic SAG (SAGA)*: Let $(\tau_{i,k})_{i \in [N], k \geq 0}$ be defined as (3), pick $i_k$ essentially cyclicly, then update $x^{k+1}$ as $\begin{cases} v^{k+1} = v^k - \nabla f_{i_k}(x^{k-\tau_{i_k,k}}) + \nabla f_{i_k}(x^k), \\ \quad x^{k+1} = \mathbf{prox}_{\gamma_k g}(x^k - \gamma_k v^{k+1}). \end{cases}$

E4. *Deterministic SVRG*: Pick $i_k$ cyclicly, i.e., $i_k \equiv k \pmod m$, let $t = \lfloor \frac{k}{m} \rfloor$, take $\tilde{x}$ being any one from $\{x^{mt}, x^{mt+1}, \ldots, x^{mt+t-1}\}$. And then update $x^{k+1}$ as $x^{k+1} = \mathbf{prox}_{\gamma_k g}[x^k - \gamma(\nabla f_{i_k}(x^k) - \nabla_k f_{i_k}(\tilde{x}) + \nabla f(\tilde{x}))]$.

E5. *Decentralized Parallel Stochastic Gradient Descent (DPSGD)*: The algorithm is proposed in [17] to solve $\min_X \{D(X) := \sum_{j=1}^{n} \sum_{i \in S_j} f_i(x_j) + \frac{1}{2} \|(I-W)^{\frac{1}{2}} X\|_F^2\}$, where $X := [x_1, x_2, \ldots, x_n]^\top$, $W$ is mixing matrix, and $S_j$ is the neighbour set of $j$. In each iteration, the DPSGD computes a stochastic gradient of $\sum_{i \in S_j} f_i(x_j)$ at node $j$, and then computes the neighborhood weighted average to update the local variable.

E6. *Forward-backward splitting by Parameter Server Computing:* The forward-backward splitting for problem (1) can be implemented on a parameter server computing model, in which, the worker nodes communicate synchronously with the parameter server but not directly with each other. Node $i$ just computes $\nabla_i f(\hat{x}^k)$ and sends the result to the parameter server, where $\hat{x}^k$ comes from the shared memory with bounded delay $x^k$. In the parameter server, the gradients are collected and the iterate is updated (implement the proximal map computation). The algorithm is a special case of the general PIAG. More details about parameter server computing can be found in [24].

E7. *LAG: Lazily aggregated gradient:* This algorithm is also designed with parameter server computing. Different from E.7, the main motivation of this algorithm is to reduce the communications. In this setting, the parameter server broadcasts the current iteration $x^k$ to the workers which is low-costly; while the data transition cost from the worker to parameter server is high. In this case, authors in [9] propose LAG whose core idea is disabling the data feedback from worker to parameter server if the gradients change slightly. It is not difficult to verify that the general PIAG contains LAG.

## 1.4   Contribution

In the perspective of algorithms, this paper proposes the general PIAG, which not only covers various classical PIAG-like algorithms including the inexact schemes but also derives novel algorithms. In the perspective of theory, we build better and novel results compared with previous literature. Specifically, the contribution of this paper can be summarized as follows:

**I. General scheme:** We propose a general PIAG algorithm, which covers various classical algorithms in network optimization, distributed optimization, machine learning areas. We unify all these algorithms into one mathematical scheme.

**II. Novel algorithm:** We apply the line search strategy to PIAG and prove its convergence. The numerical results demonstrate its efficiency.

**III. Novel proof techniques:** Compared with previous convergence analysis of PIAG, we use a new proof technique: Lyapunov function analysis. Due to this, we can build much stronger theoretical results with more general scheme. The Lyapunov function analysis is through the paper for both convex and nonconvex cases.

**IV. Better theoretical results:** The previous convergence results of PIAG is restricted to strongly convex case and the stepsize depends on the strong convexity constant. We get rid of this constant and still guarantees the linear convergence with much larger stepsizes, even under a weaker assumption. For the cyclic PIAG, the stepsize can be half of the gradient descent algorithm.

**V. Novel theoretical results:** Many new results are proved in this paper. We list them as follows:

- V.1. The convergence of nonconvex PIAG is studied. And in the expectation-free case, the sequence convergence is proved under the semi-algebraic property.

Table 1: Contributions of this paper

| Novel results | | |
|---|---|---|
| Algorithms | Theory | |
| 1. A general scheme which covers various classical algorithms. 2. Line search of PIAG is proposed and analyzed. | Theoretical results | 1.The (in)exact convergence of nonconvex PIAG is studied. The sequence convergence is proved by means of semi-algebraic property. 2. Sublinear convergence of (in)exact PIAG under general convex assumptions. |
| | Proof technique | Lyapunov function analysis |
| Better results | 1. Much larger stepsize is proved to be convergence. 2. The numerical results show that line search of PIAG performs much better than PIAG. | |

- V.2. The convergence of the inexact PIAG is proved for both convex and nonconvex cases. In the convex case, the convergence rates are exploited if the noises are promised to follow certain assumptions. In the nonconvex case, the sequence convergence is also prove under semi-algebraic property and assumption on the noises.

- V.3. We proved the sublinear convergence of PIAG under general convex assumptions. To the best of our knowledge, it is the first time to prove the non-ergodic $O(1/k)$ convergence rate of PIAG. And, we also proved the non-ergodic $O(1/k)$ convergence rate for inexact PIAG.

- V.4. The convergence of line search of PIAG is proved for both convex and nonconvex cases. The convergence rates are also presented in the convex case.

## 2   Preliminaries

Through the paper, we use the notation $\Delta^k := x^{k+1} - x^k$, and $\sigma_k := \left[ \mathbb{E}(\|v^k - \sum_{i=1}^m \nabla f_i(x^{k-\tau_{i,k}})\|^2 \mid \chi^k) \right]^{\frac{1}{2}}$. Assume that $f_i$ is differentiable and $\nabla f_i$ is $L_i$-Lipschitz continuous. Then, $\nabla f$ is Lipschitz continuous with $L := \sum_{i=1}^m L_i$. The maximal delay is $\tau := \max_{i,k}\{\tau_{i,k}\}$. The convergence analysis in the paper depends on the square summable assumption on $(\sigma_k)_{k\geq 0}$, i.e., $\sum_i \sigma_i^2 < +\infty$. That is why the general PIAG just contains deterministic SAGA and SVRG, in which case $\sigma_k \equiv 0$. However, the SAGA and SVRG may not have the summable assumption held. In the deterministic case, $\sigma_k = \|e^k\|$, according to the general PIAG we defined in (2). Then we only need $\sum_i \|e^i\|^2 < +\infty$. Further, if the noise vanishes, the assumption certainly holds. Besides the deterministic case we discussed above, the stochastic coordinate descent algorithm (with asynchronous parallel) can also satisfy this assumption. Taking the stochastic coordinate descent algorithm for example, in this algorithm, $\sigma_k^2 = \frac{N-1}{N}\mathbb{E}\|\nabla f(x^k)\|^2 = (N-1)\mathbb{E}\|\Delta^k\|^2$. In the stochastic coordinate descent algorithm, it is easy to prove $\sum_i \mathbb{E}\|\Delta\|^2 < +\infty$ if the stepsize is well chosen. For the asynchronous parallel algorithm, by assuming the independence between $\hat{x}^k$ with $i_k$, we can prove the same result given in [Lemma 1, [27]]. We introduce the definitions of subdifferentials. The details can be found in [19, 22, 23].

**Definition 1** *Let $J : \mathbb{R}^N \to (-\infty, +\infty]$ be a proper and lower semicontinuous function. The subdifferential, of $J$ at $x \in \mathbb{R}^N$, written as $\partial J(x)$, is defined as*

$$\partial J(x) := \{u \in \mathbb{R}^N : \exists\, x^k \to x,\ u^k \to u,\ such\ that \lim_{\substack{y \neq x^k \\ y \to x^k}} \inf \frac{J(y) - J(x^k) - \langle u^k, y - x^k \rangle}{\|y - x^k\|_2} \geq 0 \}.$$

## 3   Convergence analysis

The analysis in this section is heavily based on the following Lyapunov function:

$$\xi_k(\varepsilon, \delta) := F(x^k) + \frac{L}{2\varepsilon} \sum_{d=k-\tau}^{k-1} (d - (k-\tau) + 1)\|\Delta^d\|^2 + \frac{1}{2\delta} \sum_{i=k}^{+\infty} \sigma_i^2 - \min F, \qquad (4)$$

where $\varepsilon, \delta > 0$ will be determined later, based on the step size $\gamma$ and $\tau$ (the bound for $\tau_{i,k}$). We discuss the convergence when $g$ (the regularized function in (1)) is convex or nonconvex separately. The main difference of the two cases is the upper bound of the stepsize. Due to the convexity of $g$, the upper bound of the stepsize in the first case is twice as the second one.

## 3.1 $g$ is convex

When $g$ is convex, we consider three different types of convergence: the first one is in the perspective of expectation, the second one is about almost surely convergence, while the last one considers the semi-algebraic property[18, 15, 7].

**Convergence in the perspective of expectation:**

**Lemma 1** *Let $f$ be a function (may be nonconvex) with $L$-Lipschitz gradient and $g$ is convex, and finite $\min F$. Let $(x^k)_{k \geq 0}$ be generated by the general PIAG, and $\max_{i,k}\{\tau_{i,k}\} \leq \tau$, and $\sum_i \sigma_i^2 < +\infty$. Choose the step size $\gamma_k \equiv \gamma = \frac{2c}{(2\tau+1)L}$ for arbitrary fixed $0 < c < 1$. Then we can choose $\varepsilon, \delta > 0$ to obtain*

$$\mathbb{E}\xi_k(\varepsilon, \delta) - \mathbb{E}\xi_{k+1}(\varepsilon, \delta) \geq \frac{1}{4}\left(\frac{1}{\gamma} - \frac{L}{2} - \tau L\right) \cdot \mathbb{E}\|\Delta^k\|^2, \quad \lim_k \mathbb{E}\|\Delta^k\| = 0. \tag{5}$$

With the Lipschitz continuity of $f$, we are prepared to present the convergence result.

**Theorem 1** *Assume the conditions of Lemma 1 hold and $\sum_i \sigma_i^2 < +\infty$, and $(x^k)_{k \geq 0}$ is generated by general PIAG. Then, we have $\lim_k \mathbb{E}[\mathrm{dist}(\boldsymbol{0}, \partial F(x^k))] = 0$.*

**Remark 1** *For the cyclic PIAG, $\tau = M$. If we apply the gradient descent for (1), the stepsize should be $0 < \gamma < \frac{2c}{ML}$, for some $0 < c < 1$. In this case, the stepsize of cyclic PIAG is the half of the gradient descent algorithm for this problem.*

**Convergence in the perspective of almost surely:** The almost surely convergence is proved in this part. We consider a Lyapunov function which is modification of (4) as

$$\hat{\xi}_k(\varepsilon, \delta) := F(x^k) + \kappa \cdot \sum_{d=k-\tau}^{k-1} (d - (k - \tau) + 1)\|\Delta^d\|^2 + \frac{1}{2\delta} \sum_{i=k}^{+\infty} \sigma_i^2 - \min F, \tag{6}$$

where we assume $\tau \geq 1$ and

$$\kappa := \frac{L}{2\varepsilon} + \frac{1}{4\tau}\left(\frac{1}{\gamma} - \frac{L}{2} - \tau L\right). \tag{7}$$

A lemma on *nonnegative almost supermartingales* [21], whose details are included in the appendix, is needed to prove the almost sure convergence.

**Theorem 2** *Assume the conditions of Lemma 1 hold and $\sum_i \sigma_i^2 < +\infty$, and $(x^k)_{k \geq 0}$ is generated by general PIAG. Then we have $\mathrm{dist}(\boldsymbol{0}, \partial F(x^k)) \to 0, a.s.$*

**Convergence under semi-algebraic property:** If the function $F$ satisfies the semi-algebraic property[3], we can obtain more results for the inexact proximal incremental aggregated gradient algorithm. In this case, the expectation of (24) and (28) can both be removed. Similar to [Theorem 1, [28]], we can derive the following result.

**Theorem 3** *Assume the conditions of Lemma 1 hold, and $F$ satisfies the semi-algebraic property, and $(x^k)_{k \geq 0}$ is generated by (in)exact PIAG, and $\|e^k\| \sim O(\frac{1}{k^\eta})$ ($\eta > 1$), then, $(x^k)_{k \geq 0}$ converges to a critical point of $F$.*

## 3.2 $g$ is nonconvex

In this subsection, we consider the case when $g$ is nonconvex. Under this weaker assumption, the stepsizes are reduced for the convergence. Like previous subsection, we also consider three kinds of convergence. We list them as sequence.

**Proposition 1** *Assume the conditions of Theorem 1 hold except that $g$ is nonconvex and $\gamma_k \equiv \gamma = \frac{c}{(2\tau+1)L}$ for arbitrary fixed $0 < c < 1$. Then, we have $\lim_k \mathbb{E}[\text{dist}(\boldsymbol{0}, \partial F(x^k))] = 0$.*

**Proposition 2** *Assume the conditions of Proposition 1 hold, then, we have $\text{dist}(\boldsymbol{0}, \partial F(x^k)) \to 0, a.s.$*

**Proposition 3** *Assume the conditions of Theorem 3 hold except that $g$ is nonconvex and $\gamma_k \equiv \gamma = \frac{c}{(2\tau+1)L}$ for arbitrary fixed $0 < c < 1$, then, $(x^k)_{k \geq 0}$ converges to a critical point of $F$.*

## 4  Convergence rates in convex case

In this part, we prove the sublinear convergence rates of the general proximal incremental aggregated gradient algorithm under general convex case, i.e., both $f$ and $g$ are convex. The analysis in the part uses a slightly modified Lyapunov function

$$F_k(\varepsilon, \delta) := F(x^k) + \kappa \cdot \sum_{d=k-\tau}^{k-1} (d - (k - \tau) + 1)\|\Delta^d\|^2 + \lambda_k - \min F, \tag{8}$$

where $\kappa$ is given in (7) and $\lambda_k := \frac{1}{2\delta} \sum_{i=k}^{+\infty} \sigma_i^2 + \sum_{i=k}^{+\infty} \phi_i^2$ and $(\phi_k)_{k \geq 0}$ is a nonnegative sequence. Here, we assume $\tau \geq 1$.

### 4.1  Technical lemma

This part presents a technique lemma. The sublinear and linear convergence results are both derived from this lemma.

**Lemma 2** *Assume the gradient of $f$ is Lipschitz with $L$ and $g$ is convex. Choose the step size $\gamma_k \equiv \gamma = \frac{2c}{(2\tau+1)L}$ for arbitrary fixed $0 < c < 1$. For any positive sequence $(\phi_k)_{k \geq 0}$ satisfying $\frac{\sigma_k}{\sqrt{2}} \leq \phi_k, \sum_{i=k}^{+\infty} \phi_i^2 \leq D\phi_k^2$, for some $D > 0$. Let $\overline{x^k}$ denote the projection of $x^k$ to $\arg \min F$, assumed to exist, and let*

$$\begin{cases} \alpha & := \max\{\frac{1}{\gamma} + L + \kappa\tau, 2D\} / [\min\{\frac{L}{8\tau}(\frac{1}{\gamma} - \frac{1}{2} - \tau), 1\}] \\ \beta & := (\tau + 1)(\frac{1}{\gamma} + L) + 1 \end{cases}.$$

*Then, there exist $\varepsilon, \delta > 0$ such that:*

$$(\mathbb{E}F_{k+1}(\varepsilon, \delta))^2 \leq \alpha(\mathbb{E}F_k(\varepsilon, \delta) - \mathbb{E}F_{k+1}(\varepsilon, \delta)) \times (\kappa\tau \sum_{d=k-\tau}^{k-1} \mathbb{E}\|\Delta^d\|^2 + \beta\mathbb{E}\|x^{k+1} - \overline{x^{k+1}}\|^2 + \lambda_k). \tag{9}$$

### 4.2  Sublinear convergence rate under general convexity

In this subsection, we present the sublinear convergence of the general proximal incremental aggregated gradient algorithm.

**Theorem 4** *Assume the gradient of $f$ is Lipschitz continuous with $L$ and $g$ is convex, and $\boldsymbol{prox}_g(\cdot)$ is bounded. Choose the step size $\gamma_k \equiv \gamma = \frac{2c}{(2\tau+1)L}$ for arbitrary fixed $0 < c < 1$. Let $(x^k)_{k \geq 0}$ be generated by the general proximal incremental aggregated gradient algorithm. And the $\sigma_k \sim O(\zeta^k)$, where $0 < \zeta < 1$. Then, we have*

$$\mathbb{E}F(x^k) - \min F \sim O(\frac{1}{k}). \tag{10}$$

In many cases, $\boldsymbol{prox}_g(\cdot)$ may be unbounded. However, we can slightly modified the algorithm. For example, in the LASSO problem

$$\min_x \{\|b - Ax\|_2^2 + \|x\|_1\}, \tag{11}$$

we can easily see that $\|x^*\|_1 \leq \|b - A \cdot \mathbf{0}\|_2^2 + \|\mathbf{0}\|_1 = \|b\|_2^2$. That means the solution set of (11) is bounded by $X := [-\|b\|_2^2, \|b\|_2^2]^N$. Then, we can turn to solve $\min_x\{\|b - Ax\|_2^2 + \|x\|_1 + \delta_X(x)\}$. And we can set $g(\cdot) = \|\cdot\|_1 + \delta_X(\cdot)$ rather than $\|\cdot\|_1$. Luckily, the proximal map of $\|\cdot\|_1 + \delta_X(\cdot)$ is proximable. With [Theorem 2, [31]], we have $\left[\mathbf{prox}_{\|\cdot\|_1+\delta_X(\cdot)}(x)\right]_i = \mathbf{prox}_{\delta_{[-\|b\|_2^2, \|b\|_2^2]}}[\mathbf{prox}_{|\cdot|}(x_i)]$ for $i \in [1, 2, \ldots, N]$. In the deterministic case, the sublinear convergence still holds even the $\mathbf{prox}_g(\cdot)$ is unbounded. The boundedness of the $\mathbf{prox}_g(\cdot)$ is used to derive the boundedness of sequence $(x^k)_{k \geq 0}$. In fact, this boundedness can be obtained by the coercivity of function $F$ in the deterministic case.

**Proposition 4** *Assume the condition of Theorem 4 hold. Let $(x^k)_{k \geq 0}$ be generated by the (in)exact PIAG, then $F(x^k) - \min F \sim O(\frac{1}{k})$.*

To the best of our knowledge, this is the first time to prove the sublinear convergence rate for the proximal incremental aggregated gradient algorithm.

### 4.3 Linear convergence with larger stepsize

Assume that the function $F$ satisfies the following condition

$$F(x) - \min F \geq \nu\|x - \overline{x}\|^2, \tag{12}$$

where $\overline{x}$ is the projection of $x$ to the set $\arg\min F$, and $\nu > 0$. This property is weaker than the strongly convexity. If $F$ is further differentiable, condition (12) is equivalent to the *restricted strongly convexity* [16].

**Theorem 5** *Assume the gradient of $f$ is Lipschitz with $L$ and $g$ is convex, and the function $F$ satisfies condition (12). Choose the step size $\gamma_k \equiv \gamma = \frac{2c}{(2\tau+1)L}$ for arbitrary fixed $0 < c < 1$. And the $\sigma_k \sim O(\zeta^k)$, where $0 < \zeta < 1$. Then, we have*

$$\mathbb{E}F(x^k) - \min F \sim O(\omega^k), \tag{13}$$

*for some $0 < \omega < 1$.*

Compared with the existing linear convergence results in [30, 12], our theoretical findings enjoys two advantages: 1. we generalize the strong convexity to a much weaker condition (12); 2. the stepsize gets rid of the parameter $\nu$ which promises larger descent of the algorithm when $\nu$ is small.

## 5 Line search of the proximal incremental gradient algorithm

In this part, we consider a line search version of the deterministic proximal incremental gradient algorithm. First, we set $\gamma_k \equiv \frac{c}{(2\tau+1)L}$ if $g$ is nonconvex, and $\gamma_k \equiv \frac{2c}{(2\tau+1)L}$ if $g$ is convex. The scheme of the algorithm can be presented as follows: **Step 1** Compute the point $v^k = \sum_{i=1}^m \nabla f_i(x^{k-\tau_{i,k}})$. **Step 2** Find $j_k$ as the smallest integer number $j$ which obeys that $y^k = \mathbf{prox}_{\eta^{j_k}c_1 g}[x^k - \eta^{j_k}c_1 v^k]$. and $\langle v^k, y^k - x^k\rangle + g(y^k) - g(x^k) \leq -\frac{c_2}{2}\|y^k - x^k\|^2$ where $0 < \eta < 1$ and $c_1, c_2 > 0$ the parameters. Set $\eta_k = \eta^{j_k}c_1$ if $\eta_k \geq \gamma$ and $\eta_k = \gamma$ if else. The point $x^{k+1}$ is generated by $x^{k+1} = \mathbf{prox}_{\eta_k g}[x^k - \eta_k v^k]$.

Without the noise, the Lyapunov function can get one parameter free in the analysis (we can get rid of $t$). Thus, the Lyapunov function used in this part can be described as

$$\xi_k(\varepsilon) := F(x^k) + \frac{L}{2\varepsilon}\sum_{d=k-\tau}^{k-1}(d - (k-\tau) + 1)\|\Delta^d\|^2 - \min F. \tag{14}$$

**Lemma 3** *Let $f$ be a function (may be nonconvex) with $L$-Lipschitz gradient and $g$ is nonconvex, and finite $\min F$. Let $(x^k)_{k \geq 0}$ be generated by the proximal incremental aggregated gradient algorithm with line search, and $\max_{i,k}\{\tau_{i,k}\} \leq \tau$. Choose the parameter $c_2 \geq \frac{(2\tau+1)L}{c}$ and $0 < c < 1$. It then holds that $\lim_k \text{dist}(\mathbf{0}, \partial F(x^k)) = 0$.*

In previous result, if $g$ is convex, the lower bound of $c_2$ can be shortened by half. This is because (68) in the *Appendix* can be improved as $F(x^{k+1}) - F(x^k) \leq \frac{L}{2\varepsilon} \sum_{d=k-\tau}^{k-1} \|\Delta^d\|^2 + \left[\frac{(\tau\varepsilon+1)L}{2} - \frac{(2\tau+1)L}{c}\right] \|\Delta^k\|^2$. This result is proved by (21). Thus, we can obtain the following result.

**Lemma 4** *Assume conditions of Lemma 3 hold except that both $f$ and $g$ are convex and $c_2 \geq \frac{(2\tau+1)L}{2c}$. It then holds that $\lim_k \mathrm{dist}(\boldsymbol{0}, \partial F(x^k)) = 0$.*

In fact, we can also derive the convergence rate for the line search version in the convex case. The proof is very similar to the one in Section 4. Thus, we just present the sketch. Like the previous analysis, a modified Lyapunov function is needed $F_k(\varepsilon) := F(x^k) + \tilde{\kappa} \cdot \sum_{d=k-\tau}^{k-1}(d - (k-\tau) + 1)\|\Delta^d\|_2^2 - \min F$, where $\tilde{\kappa} := \frac{L}{2\varepsilon} + \frac{1-c}{8c\tau}(L + 2\tau L)$. With this Lyapunov function and suitable $\varepsilon$, we prove the following two inequalities

$$F_k(\varepsilon) - F_{k+1}(\varepsilon) \geq \min\{\frac{1-c}{8c\tau}(L + 2\tau L), 1\} \cdot (\sum_{d=k-\tau}^{k} \|\Delta^d\|^2), \qquad (15)$$

and

$$(F_{k+1}(\varepsilon))^2 \leq (\frac{1}{\gamma} + L + \tilde{\kappa}\tau) \times (\sum_{d=k-\tau}^{k} \|\Delta^d\|^2)$$

$$\times \left([(\tau + 1)(\frac{1}{\gamma} + L) + 1]\|x^{k+1} - \overline{x^{k+1}}\|^2 + \tilde{\kappa}\tau \sum_{d=k-\tau}^{k-1} \|\Delta^d\|^2\right). \qquad (16)$$

With (15) and (16), we then derive the following lemma.

**Theorem 6** *Let $f$ be a convex function with $L$-Lipschitz gradient and $g$ is convex, and finite $\min F$. Let $(x^k)_{k \geq 0}$ be generated by the proximal incremental aggregated gradient algorithm with line search, and $\max_{i,k}\{\tau_{i,k}\} \leq \tau$. Choose the parameter $c_2 \geq \frac{(2\tau+1)L}{2c}$ and $0 < c < 1$. Then, there exist $\varepsilon > 0$ such that:*

$$(F_{k+1}(\varepsilon))^2 \leq \tilde{\alpha}(F_k(\varepsilon) - F_{k+1}(\varepsilon)) \times (\tilde{\kappa}\tau \sum_{d=k-\tau}^{k-1} \|\Delta^d\|^2 + \beta\|x^{k+1} - \overline{x^{k+1}}\|^2), \qquad (17)$$

*where $\tilde{\alpha} := (\frac{1}{\gamma} + L + \tilde{\kappa}\tau)/[\min\{\frac{1-c}{8c\tau}(L + 2\tau L), 1\}]$. Further more, if $F$ is coercive, $F(x^k) - \min F \sim O(\frac{1}{k})$. If $F$ satisfies condition (12), $F(x^k) - \min F \sim O(\tilde{\omega}^k)$ for some $0 < \tilde{\omega} < 1$.*

## 6   Numerical results

Now we use some numerical experiments to show how the line search strategy can accelerate the PIAG algorithms. Here we considered the following two updating rules,

1. *scheme I*: $x^{k+1} = \mathbf{prox}_{\gamma g}[x^k - \gamma(w_j^{k+1} - w_j^k + \sum_{i=1}^{m} w_i^k)]$,
2. *scheme II*: $x^{k+1} = \mathbf{prox}_{\gamma g}[x^k - \gamma(w_j^{k+1} - w_j^{mt} + \sum_{i=1}^{m} w_i^{mt})]$,

where $j \equiv k (\mathrm{mod}\ m), w_j^{k+1} = \nabla f_j(x^k), t = \lfloor \frac{k}{m} \rfloor$. We tested binary classifiers on MNIST, ijcnn1. To include all convex and nonconvex cases, we choose logistic regression (convex) and squared logistic loss (nonconvex) for $f$, $\ell_1$ regularization (convex) and MCP (nonconvex) for $g$. The results when using scheme I and II with and without line search are shown in Figure 6. In our experiments, we choose $\gamma = \frac{2c}{(2\tau+1)L}$ when $g$ is convex and $\frac{c}{(2\tau+1)L}$ when $g$ is nonconvex, $c = 0.99$, $c_2 = \frac{1}{\gamma}$. Our numerical results shows that the line search strategy can speed up the PIAG algorithm a lot.

## 7   Conclusion

In this paper, we consider a general proximal incremental aggregated gradient algorithm and prove several novel results. Much better results are proved under more general conditions. The core of the analysis is using the Lyapunov function analysis. We also consider the line search of proximal incremental aggregated gradient algorithm and the convergence rate is proved.

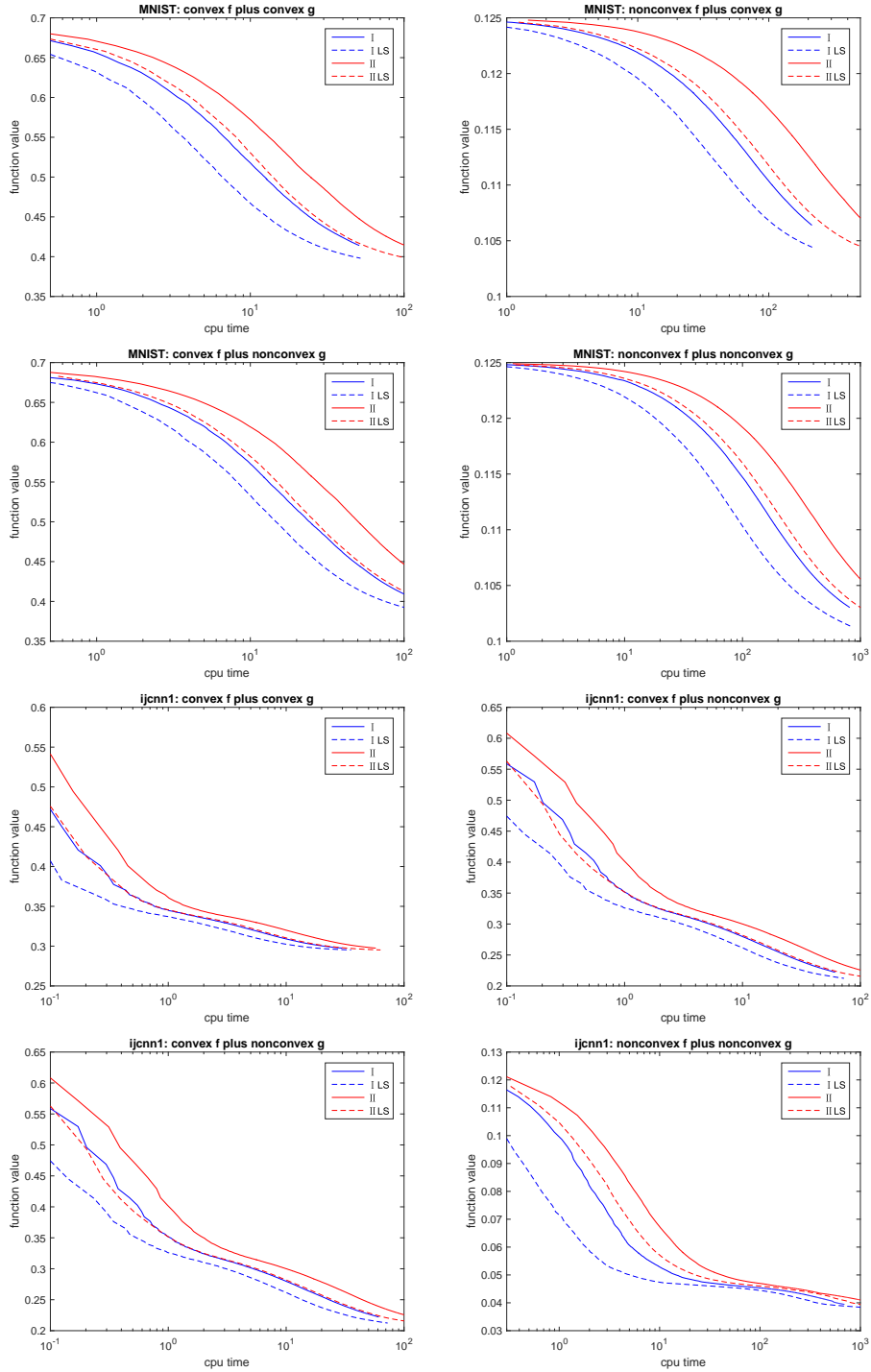

## Footnotes

[3]Semi-algebraic property used in the nonconvex optimization, more details can be found in [8, 2].

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
