[Supplementary Material · incre_supp.pdf]



# Supplementary materials for
## *General Proximal Incremental Aggregated Gradient Algorithms: Better and Novel Results under General Scheme*

348 **Lemma 5** *Let $\mathscr{F} = (\mathcal{F}^k)_{k \geq 0}$ be a sequence of sub-sigma algebras of $\mathcal{F}$ such that $\forall k \geq 0$, $\mathcal{F}^k \subset$*
349 *$\mathcal{F}^{k+1}$. Define $\ell_+(\mathscr{F})$ as the set of sequences of $[0, +\infty)$-valued random variables $(\xi_k)_{k \geq 0}$, where $\xi_k$*
350 *is $\mathcal{F}^k$ measurable, and $\ell_+^1(\mathscr{F}) := \{(\xi_k)_{k \geq 0} \in \ell_+(\mathscr{F}) | \sum_k \xi_k < +\infty \text{ a.s.}\}$. Let $(\alpha_k)_{k \geq 0}, (v_k)_{k \geq 0} \in$*
351 *$\ell_+(\mathscr{F})$, and $(\eta_k)_{k \geq 0}, (\xi_k)_{k \geq 0} \in \ell_+^1(\mathscr{F})$ be such that*

$$\mathbb{E}(\alpha_{k+1} | \mathcal{F}^k) + v_k \leq (1 + \xi_k)\alpha_k + \eta_k.$$

352 *Then $(v_k)_{k \geq 0} \in \ell_+^1(\mathscr{F})$ and $\alpha_k$ converges to a $[0, +\infty)$-valued random variable a.s..*

353 **Proof of Lemma 1**

354 Updating $x^{k+1}$ directly gives

$$\frac{x^k - x^{k+1}}{\gamma} - v^k \in \partial g(x^{k+1}). \tag{18}$$

355 With the convexity of $g$, we have

$$g(x^{k+1}) - g(x^k) \leq \langle \frac{\Delta^k}{\gamma} + v^k, -\Delta^k \rangle \tag{19}$$

356 With Lipschitz continuity of $\nabla f$,

$$f(x^{k+1}) - f(x^k) \leq \langle \nabla f(x^k), \Delta^k \rangle + \frac{L}{2}\|\Delta^k\|^2. \tag{20}$$

357 Combining (19) and (20),

$$F(x^{k+1}) - F(x^k) \overset{(19)+(20)}{\leq} \langle \nabla f(x^k) - v^k, \Delta^k \rangle + (\frac{L}{2} - \frac{1}{\gamma})\|\Delta^k\|^2$$

$$= \underbrace{\langle \nabla f(x^k) - \sum_{i=1}^m \nabla f_i(x^{k-\tau_{i,k}}), \Delta^k \rangle}_{I} + (\frac{L}{2} - \frac{1}{\gamma})\|\Delta^k\|^2 + \underbrace{\langle \sum_{i=1}^m \nabla f_i(x^{k-\tau_{i,k}}) - v^k, \Delta^k \rangle}_{II}. \tag{21}$$

358 In the following, we give bounds of $I, II$ (or expectation). By Cauchy's inequality with $\delta$,

$$II \leq \frac{\|v^k - \sum_{i=1}^m \nabla f_i(x^{k-\tau_{i,k}})\|^2}{2\delta} + \frac{\delta}{2}\|\Delta^k\|^2. \tag{22}$$

359 On the other hand,

$$I \overset{a)}{\leq} \sum_{i=1}^m L_i \|x^k - x^{k-\tau_{i,k}}\| \cdot \|\Delta^k\|$$

$$\overset{b)}{\leq} \sum_{i=1}^m L_i \left( \sum_{d=k-\tau}^{k-1} \|\Delta^d\| \right) \cdot \|\Delta^k\|$$

$$\overset{c)}{\leq} L \sum_{d=k-\tau}^{k-1} \|\Delta^d\| \cdot \|\Delta^k\|$$

$$\overset{d)}{\leq} \frac{L}{2\varepsilon} \sum_{d=k-\tau}^{k-1} \|\Delta^d\|^2 + \frac{\tau\varepsilon L}{2}\|\Delta^k\|^2. \tag{23}$$

360 where $a)$ depends on the Lipschitz continuity of $\nabla f_i$, $b)$ is obtained from the inequality $\|x^k -$
361 $x^{k-\tau_{i,k}}\| \leq \sum_{d=k-\tau_{i,k}}^{k-1} \|\Delta^d\| \leq \sum_{d=k-\tau}^{k-1} \|\Delta^d\|$, $c)$ is due to the fact $L = \sum_{i=1}^m L_i$, and $d)$ uses

Cauchy's inequality with $\varepsilon \|\Delta^d\| \cdot \|\Delta^k\| \leq \frac{1}{2\varepsilon}\|\Delta^d\|^2 + \frac{\varepsilon}{2}\|\Delta^k\|^2$. Combining (21), (22) and (23) and taking conditional expectation over $\chi^k$, we have

$$
\mathbb{E}(F(x^{k+1}) \mid \chi^k) - F(x^k) \leq \frac{L}{2\varepsilon} \sum_{d=k-\tau}^{k-1} \|\Delta^d\|^2
$$
$$
+ \left[ \frac{(\tau\varepsilon+1)L}{2} - \frac{1}{\gamma} + \frac{\delta}{2} \right] \mathbb{E}(\|\Delta^k\|^2 \mid \chi^k) + \frac{\sigma_k^2}{2\delta}. \tag{24}
$$

If $\gamma < \frac{2}{(2\tau+1)L}$, we can choose $\varepsilon, \delta > 0$ such that

$$
\varepsilon + \frac{1}{\varepsilon} = 1 + \frac{1}{\tau}(\frac{1}{\gamma L} - \frac{1}{2}), \delta = \frac{1}{2}(\frac{1}{\gamma} - \frac{L}{2} - \tau L)
$$

Then, with direct calculations and substitutions, we have:

$$
\xi_k(\varepsilon, \delta) - \mathbb{E}(\xi_{k+1}(\varepsilon, \delta) \mid \chi^k) \overset{(4)}{=} F(x^k) - \mathbb{E}(F(x^{k+1}) \mid \chi^k)
$$
$$
+ \frac{L}{2\varepsilon} \sum_{d=k-\tau}^{k-1} (d - (k - \tau) + 1)\|\Delta^d\|^2
$$
$$
- \frac{L}{2\varepsilon} \sum_{d=k+1-\tau}^{k-1} (d - (k - \tau))\|\Delta^d\|^2 - \frac{L}{2\varepsilon}\tau\mathbb{E}(\|\Delta^k\|^2 \mid \chi^k) + \frac{\sigma_k^2}{2\delta}
$$
$$
\overset{c)}{=} F(x^k) - \mathbb{E}(F(x^{k+1}) \mid \chi^k) + \frac{L}{2\varepsilon} \sum_{d=k-\tau}^{k-1} \|\Delta^d\|^2 - \frac{L}{2\varepsilon}\tau\mathbb{E}(\|\Delta^k\|^2 \mid \chi^k) + \frac{\sigma_k^2}{2\delta}
$$
$$
\overset{(24)}{\geq} \frac{1}{4}(\frac{1}{\gamma} - \frac{L}{2} - \tau L) \cdot \mathbb{E}(\|\Delta^k\|^2 \mid \chi^k), \tag{25}
$$

where c) follows from $(d-(k-\tau)+1)\|\Delta^d\|^2 - (d-(k-\tau))\|\Delta^d\|^2 = \|\Delta^d\|^2$. Taking expectation on both sides of (25), we then prove the result. Therefore we have $\mathbb{E}\|\Delta^k\|^2 \in \ell^1$ by using a telescoping sum[1].

**Proof of Theorem 1**

Obviously, we have
$$
(x^k - x^{k+1})/\gamma - v^k \in \partial g(x^{k+1}). \tag{26}
$$

That means
$$
(x^k - x^{k+1})/\gamma + \nabla f(x^{k+1}) - v^k \in \nabla f(x^{k+1}) + \partial g(x^{k+1}) = \partial F(x^{k+1}). \tag{27}
$$

Thus, we have
$$
\mathbb{E}[\text{dist}(\mathbf{0}, \partial F(x^{k+1}))]
$$
$$
\leq \mathbb{E}\|(x^k - x^{k+1})/\gamma + \nabla f(x^{k+1}) - \sum_{i=1}^m \nabla f_i(x^{k-\tau_{i,k}}) + \sum_{i=1}^m \nabla f_i(x^{k-\tau_{i,k}}) - v^k\|
$$
$$
\leq \mathbb{E}\|\Delta^k\|/\gamma + L \sum_{d=k-\tau}^k \mathbb{E}\|\Delta^d\| + \sigma_k. \tag{28}
$$

Taking the limitation $k \to +\infty$ and with Lemma 1, the result is then proved.

**Proof of Theorem 2**

Since $0 < \gamma < \frac{2}{2\tau+1}$, we can choose $\varepsilon, \delta > 0$ such that
$$
\varepsilon + \frac{1}{\varepsilon} = 1 + \frac{1}{\tau}(\frac{1}{\gamma} - \frac{1}{2}), \delta = \frac{L}{4\tau}(\frac{1}{\gamma} - \frac{1}{2} - \tau). \tag{29}
$$

375 With direct computations, we have

$$\hat{\xi}_k(\varepsilon, \delta) - \mathbb{E}[\hat{\xi}_k(\varepsilon, \delta) \mid \chi^k] + \frac{\delta}{2}\mathbb{E}(\|\Delta^k\|^2 \mid \chi^k)$$

$$\overset{a)}{\geq} F(x^k) - \mathbb{E}[F(x^{k+1}) \mid \chi^k] + \kappa \sum_{d=k-\tau}^{k-1}(d - (k-\tau) + 1)\mathbb{E}\|\Delta^d\|^2$$

$$- \kappa \sum_{d=k+1-\tau}^{k-1}(d - (k-\tau))\mathbb{E}\|\Delta^d\|^2 - \kappa\tau\mathbb{E}\|\Delta^k\|^2 + \frac{\sigma_k^2}{2\delta} + \frac{\delta}{2}\mathbb{E}(\|\Delta^k\|^2 \mid \chi^k)$$

$$= F(x^k) - \mathbb{E}[F(x^{k+1}) \mid \chi^k] + \sum_{d=k-\tau}^{k-1}\kappa\|\Delta^d\|^2$$

$$- \kappa\tau\mathbb{E}(\|\Delta^k\|^2 \mid \chi^k) + \frac{\sigma_k^2}{2\delta} + \frac{\delta}{2}\mathbb{E}(\|\Delta^k\|^2 \mid \chi^k)$$

$$\overset{b)}{\geq} (\kappa - \frac{L}{2\varepsilon}) \cdot \Big( \sum_{d=k-\tau}^{k-1} \|\Delta^d\|^2 \Big) + \Big[\frac{1}{\gamma} - \frac{(\tau\varepsilon + 1)L}{2} - \kappa\tau\Big]\mathbb{E}(\|\Delta^k\|^2 \mid \chi^k)$$

$$\overset{c)}{=} \frac{1}{4\tau}(\frac{1}{\gamma} - \frac{L}{2} - \tau L) \cdot \Big( \sum_{d=k-\tau}^{k-1}\|\Delta^d\|^2 \Big) + \frac{1}{4}(\frac{1}{\gamma} - \frac{L}{2} - \tau L)\cdot\mathbb{E}(\|\Delta^k\|^2 \mid \chi^k)$$

$$\overset{d)}{\geq} \frac{1}{4\tau}(\frac{1}{\gamma} - \frac{L}{2} - \tau L) \cdot \Big( \sum_{d=k-\tau}^{k-1}\|\Delta^d\|^2 \Big) + \frac{1}{4\tau}(\frac{1}{\gamma} - \frac{L}{2} - \tau L)\cdot\mathbb{E}(\|\Delta^k\|^2 \mid \chi^k)$$

$$= \frac{1}{4\tau}(\frac{1}{\gamma} - \frac{L}{2} - \tau L) \cdot \Big( \sum_{d=k-\tau}^{k-1}\|\Delta^d\|^2 \Big) + \frac{\delta}{2}\mathbb{E}(\|\Delta^k\|^2 \mid \chi^k), \tag{30}$$

376 where a) follows from the definition $\hat{\xi}_k(\varepsilon, \delta)$, b) is from (24), c) is a direct computation using (7) and
377 (29), d) is due to that $\mathbb{E}(\|\Delta^k\|^2 \mid \chi^k)) \geq 0$. Applying Lemma (5) to (30), we then have

$$\lim_k \|\Delta^k\| = 0, a.s. \tag{31}$$

378 Using the deterministic form of (28), we then prove the result.

### Proof of Theorem 3

380 Consider an auxiliary function

$$P(y_1, y_2, \ldots, y_{\tau+1}) = F(y_{\tau+1}) + \frac{L}{2\varepsilon}\sum_{d=1}^{\tau}d\|y_{d+1} - y_d\|^2 \tag{32}$$

381 and auxiliary point

$$y^k := (x^{k-\tau}, x^{k-\tau+1}, \ldots, x^k), \tag{33}$$

382 where $\varepsilon$ is given as the same way in Lemma 1. It is easy to see that $P$ is also semi-algebraic. With
383 (25), we can see that

$$P(y^k) - P(y^{k+1}) \geq \frac{1}{4}(\frac{1}{\gamma} - \frac{L}{2} - \tau L) \cdot \|x^{k+1} - x^k\|^2 - \frac{\sigma_k^2}{2\delta}. \tag{34}$$

384 On the other hand, with direct calculations and (28),

$$\text{dist}(\mathbf{0}, \partial P(y^{k+1})) \leq \|x^k - x^{k+1}\|/\gamma + L(\tau + 1)\sum_{d=k-\tau}^{k}\|x^{d+1} - x^d\| + \sigma_k \tag{35}$$

Thus, [(1.6), [26]] is satisfied, and with [Theorem 1, [26]], $\sum_k \|x^{k+1} - x^k\| < +\infty$. Then, $\sum_k \|y^{k+1} - y^k\| < +\infty$; that means the sequence $(y^k)_{k\geq 0}$ is convergent. Using (35), $(y^k)_{k\geq 0}$ converges to some critical point of $P$, we denote as $y^* = (y_1^*, y_2^*, \ldots, y_{\tau+1}^*)$. Noting $\text{dist}(\mathbf{0}, \partial P(y^*)) = 0$, then $y_1^* = y_2^* = \ldots = y_{\tau+1}^*$ and

$$0 = \text{dist}(\mathbf{0}, \partial F(y_1^*)) = \text{dist}(\mathbf{0}, \partial F(y_2^*)) = \ldots = \text{dist}(\mathbf{0}, \partial F(y_{\tau+1}^*)).$$

385 Thus, $(x^k)_{k\geq 0}$ converges to $y_1^* (= y_2^* = \ldots = y_{\tau+1}^*)$ which is a critical point of $F$.

## Proof of Proposition 1

We just need to prove that there exist $\varepsilon, \delta > 0$ such that

$$\mathbb{E}\xi_k(\varepsilon,\delta) - \mathbb{E}\xi_{k+1}(\varepsilon,\delta) \geq (\frac{1}{8\gamma} - \frac{L}{8} - \frac{\tau L}{4}) \cdot \mathbb{E}\|\Delta^k\|^2, \quad \lim_k \mathbb{E}\|\Delta^k\| = 0. \tag{36}$$

Updating $x^{k+1}$ directly gives

$$x^{k+1} \in \arg\min_y \{\frac{1}{2}\|x^k - \gamma v^k - y\|^2 + \gamma g(y)\}, \tag{37}$$

which directly yields

$$\frac{1}{2}\|x^k - \gamma v^k - x^{k+1}\|^2 + \gamma g(x^{k+1}) \leq \frac{1}{2}\| - \gamma v^k\|^2 + \gamma g(x^k). \tag{38}$$

After simplification, we have

$$\frac{1}{2\gamma}\|x^k - x^{k+1}\|^2 + g(x^{k+1}) \leq \langle v^k, x^k - x^{k+1}\rangle + g(x^k). \tag{39}$$

With Lipschitz continuity of $\nabla f$,

$$f(x^{k+1}) - f(x^k) \leq \langle -\nabla f(x^k), x^k - x^{k+1}\rangle + \frac{L}{2}\|x^{k+1} - x^k\|^2. \tag{40}$$

Combining (39) and (40),

$$F(x^{k+1}) - F(x^k) \overset{(39)+(40)}{\leq} \langle \nabla f(x^k) - v^k, \Delta^k\rangle + (\frac{L}{2} - \frac{1}{2\gamma})\|\Delta^k\|^2$$

$$= \underbrace{\langle \nabla f(x^k) - \sum_{i=1}^m \nabla f_i(x^{k-\tau_{i,k}}), \Delta^k\rangle}_{I} + (\frac{L}{2} - \frac{1}{2\gamma})\|\Delta^k\|^2$$

$$+ \underbrace{\langle \sum_{i=1}^m \nabla f_i(x^{k-\tau_{i,k}}) - v^k, \Delta^k\rangle}_{II} \tag{41}$$

The terms I and II has been bounded in Lemma 1. With the bounds and taking conditional expectation over $\chi^k$,

$$\mathbb{E}(F(x^{k+1}) \mid \chi^k) - F(x^k) \leq \frac{L}{2\varepsilon}\sum_{d=k-\tau}^{k-1}\|\Delta^d\|^2$$

$$+ \left[\frac{(\tau\varepsilon+1)L}{2} - \frac{1}{2\gamma} + \frac{t}{2}\right]\mathbb{E}(\|\Delta^k\|^2 \mid \chi^k) + \frac{\sigma_k^2}{2t}. \tag{42}$$

If $0 < \gamma < \frac{1}{(2\tau+1)L}$, we can choose $\varepsilon, t > 0$ such that

$$\varepsilon + \frac{1}{\varepsilon} = 1 + \frac{1}{\tau}(\frac{1}{2\gamma L} - \frac{1}{2}), t = \frac{1}{2}(\frac{1}{2\gamma} - \frac{L}{2} - \tau L).$$

Then, with direct calculations and substitutions, we have:

$$\xi_k(\varepsilon) - \mathbb{E}(\xi_{k+1}(\varepsilon) \mid \chi^k) \overset{(4)}{=} F(x^k) - \mathbb{E}(F(x^{k+1}) \mid \chi^k) + \frac{L}{2\varepsilon}\sum_{d=k-\tau}^{k-1}(d - (k-\tau) + 1)\|\Delta^d\|^2$$

$$- \frac{L}{2\varepsilon}\sum_{d=k+1-\tau}^{k-1}(d - (k-\tau))\|\Delta^d\|^2 - \frac{L}{2\varepsilon}\tau\mathbb{E}(\|\Delta^k\|^2 \mid \chi^k) + \frac{\sigma_k^2}{2t}$$

$$\overset{c)}{=} F(x^k) - \mathbb{E}(F(x^{k+1}) \mid \chi^k) + \frac{L}{2\varepsilon}\sum_{d=k-\tau}^{k-1}\|\Delta^d\|^2 - \frac{L}{2\varepsilon}\tau\mathbb{E}(\|\Delta^k\|^2 \mid \chi^k) + \frac{\sigma_k^2}{2t}$$

$$\overset{(38)}{\geq} \frac{1}{4}(\frac{1}{2\gamma} - \frac{L}{2} - \tau L) \cdot \mathbb{E}(\|\Delta^k\|^2 \mid \chi^k), \tag{43}$$

where c) follows from $(d - (k-\tau) + 1)\|\Delta^d\|^2 - (d - (k-\tau))\|\Delta^d\|^2 = \|\Delta^d\|^2$. With taking expectations on both sides of (43), we then prove the result.

 **Proof of Lemma 2**

399 Sketch of the proof: The proof consists of two steps. First, we prove

$$\mathbb{E}F_k(\varepsilon,\delta) - \mathbb{E}F_{k+1}(\varepsilon,\delta) \geq \min\{\frac{L}{8\tau}(\frac{1}{\gamma} - \frac{1}{2} - \tau), 1\} \cdot (\sum_{d=k-\tau}^{k} \mathbb{E}\|\Delta^d\|^2 + \phi_k^2), \qquad (44)$$

400 Second, we prove

$$(\mathbb{E}F_{k+1}(\varepsilon,\delta))^2 \leq \max\{\frac{1}{\gamma} + L + \kappa\tau, 2D\} \cdot (\sum_{d=k-\tau}^{k} \mathbb{E}\|\Delta^d\|^2 + \phi_k^2)$$

$$\cdot \left( [(\tau+1)(\frac{1}{\gamma} + L) + 1]\mathbb{E}\|x^{k+1} - \overline{x^{k+1}}\|^2 + \kappa\tau \sum_{d=k-\tau}^{k-1} \mathbb{E}\|\Delta^d\|^2 + \lambda_k \right). \quad (45)$$

401 Combining (44) and (45) gives us the claim in the lemma.

402 **Proof of (44)**

403 Since $0 < \gamma < \frac{2}{(2\tau+1)L}$, we can choose $\varepsilon, \delta > 0$ such that

$$\varepsilon + \frac{1}{\varepsilon} = 1 + \frac{1}{\tau}(\frac{1}{\gamma L} - \frac{1}{2}), \delta = \frac{1}{4\tau}(\frac{1}{\gamma} - \frac{L}{2} - \tau L). \qquad (46)$$

404 Direct subtraction of $F_k$ and $F_{k+1}$ yields:

$$\mathbb{E}[F_k(\varepsilon,t) - F_{k+1}(\varepsilon,t)] \overset{a)}{\geq} \mathbb{E}F(x^k) - \mathbb{E}F(x^{k+1}) + \kappa \sum_{d=k-\tau}^{k-1} (d - (k-\tau) + 1)\mathbb{E}\|\Delta^d\|^2$$

$$- \kappa \sum_{d=k+1-\tau}^{k-1} (d - (k-\tau))\mathbb{E}\|\Delta^d\|_2^2 - \kappa\tau\mathbb{E}\|\Delta^k\|^2 + \frac{\sigma_k^2}{2\delta} + \phi_k^2$$

$$= \mathbb{E}F(x^k) - \mathbb{E}F(x^{k+1}) + \sum_{d=k-\tau}^{k-1} \kappa\mathbb{E}\|\Delta^d\|^2 - \kappa\tau\mathbb{E}\|\Delta^k\|^2 + \frac{\sigma_k^2}{2\delta} + \phi_k^2$$

$$\overset{b)}{\geq} (\kappa - \frac{L}{2\varepsilon}) \cdot (\sum_{d=k-\tau}^{k-1} \mathbb{E}\|\Delta^d\|^2) + \left[\frac{1}{\gamma} - \frac{(\tau\varepsilon+1)L}{2} - \kappa\tau\right]\mathbb{E}\|\Delta^k\|^2 - \frac{\delta}{2}\mathbb{E}\|\Delta^k\|^2 + \phi_k^2$$

$$\overset{c)}{=} \frac{1}{4\tau}(\frac{1}{\gamma} - \frac{L}{2} - \tau L) \cdot (\sum_{d=k-\tau}^{k-1} \mathbb{E}\|\Delta^d\|^2) + \frac{1}{4}(\frac{1}{\gamma} - \frac{L}{2} - \tau L) \cdot \|\Delta^k\|^2 - \frac{\delta}{2}\mathbb{E}\|\Delta^k\|^2 + \phi_k^2$$

$$\overset{d)}{\geq} \frac{1}{4\tau}(\frac{1}{\gamma} - \frac{L}{2} - \tau L) \cdot (\sum_{d=k-\tau}^{k-1} \mathbb{E}\|\Delta^d\|^2) + \frac{1}{4\tau}(\frac{1}{\gamma} - \frac{L}{2} - \tau L) \cdot \mathbb{E}\|\Delta^k\|^2 - \frac{\delta}{2}\mathbb{E}\|\Delta^k\|^2 + \phi_k^2$$

$$= \min\{\frac{1}{8\tau}(\frac{1}{\gamma} - \frac{L}{2} - \tau L), 1\} \cdot (\sum_{d=k-\tau}^{k} \mathbb{E}\|\Delta^d\|^2 + \phi_k^2), \qquad (47)$$

405 where a) follows from the definition $F_k$, b) from taking expectation on both sides of (24), c) is a
406 direct computation using definition of $\lambda_k$, d) is due to $\tau \geq 1$.

407 **Proof of (45)**

408 The convexity of $g$ yields

$$g(x^{k+1}) - g(\overline{x^{k+1}}) \leq \langle \widetilde{\nabla}g(x^{k+1}), x^{k+1} - \overline{x^{k+1}} \rangle, \qquad (48)$$

409 where $\widetilde{\nabla}g(x^{k+1}) \in \partial g(x^{k+1})$ is arbitrary. With (18), we then have

$$g(x^{k+1}) - g(\overline{x^{k+1}}) \leq \langle \frac{x^k - x^{k+1}}{\gamma} - v^k, x^{k+1} - \overline{x^{k+1}} \rangle. \qquad (49)$$

410 Simiarly, we have

$$f(x^{k+1}) - f(\overline{x^{k+1}}) \leq \langle \nabla f(x^{k+1}), x^{k+1} - \overline{x^{k+1}} \rangle. \tag{50}$$

411 Summing (49) and (50) yields

$$F(x^{k+1}) - F(\overline{x^{k+1}}) \leq \sum_{i=1}^{m} \langle \nabla f_i(x^{k+1}) - \nabla f_i(x^{k-\tau_{i,k}}), x^{k+1} - \overline{x^{k+1}} \rangle$$

$$+ \langle \frac{x^k - x^{k+1}}{\gamma}, x^{k+1} - \overline{x^{k+1}} \rangle + \langle \sum_{i=1}^{m} \nabla f_i(x^{k-\tau_{i,k}}) - v^k, x^{k+1} - \overline{x^{k+1}} \rangle$$

$$\overset{a)}{\leq} \sum_{i=1}^{m} L_i \|x^{k+1} - x^{k-\tau_{i,k}}\| \cdot \|x^{k+1} - \overline{x^{k+1}}\|$$

$$+ \langle \frac{x^k - x^{k+1}}{\gamma}, x^{k+1} - \overline{x^{k+1}} \rangle + \langle \sum_{i=1}^{m} \nabla f_i(x^{k-\tau_{i,k}}) - v^k, x^{k+1} - \overline{x^{k+1}} \rangle$$

$$\overset{b)}{\leq} \sum_{i=1}^{m} L_i \left( \sum_{d=k-\tau}^{k} \|\Delta^d\| \right) \cdot \|x^{k+1} - \overline{x^{k+1}}\| + \frac{\|\Delta^k\|}{\gamma} \cdot \|x^{k+1} - \overline{x^{k+1}}\|$$

$$+ \langle \sum_{i=1}^{m} \nabla f_i(x^{k-\tau_{i,k}}) - v^k, x^{k+1} - \overline{x^{k+1}} \rangle$$

$$\overset{c)}{=} \sum_{d=k-\tau}^{k} L \|\Delta^d\| \cdot \|x^{k+1} - \overline{x^{k+1}}\| + \frac{\|\Delta^k\|}{\gamma} \cdot \|x^{k+1} - \overline{x^{k+1}}\|$$

$$+ \langle \sum_{i=1}^{m} \nabla f_i(x^{k-\tau_{i,k}}) - v^k, x^{k+1} - \overline{x^{k+1}} \rangle, \tag{51}$$

412 where $a)$ is due to the Lipschitz continuity of $\nabla_i f_i$, $b)$ depends on the fact $\|x^{k+1} - x^{k-\tau_{i,k}}\| \leq$
413 $\sum_{d=k-\tau}^{k} \|\Delta^d\|$, and $L = \sum_{i=1}^{m} L_i$. With (8) and (51), we have

$$F_{k+1}(\varepsilon, t) \leq \sum_{d=k-\tau}^{k-1} L\|\Delta^d\| \cdot \|x^{k+1} - \overline{x^{k+1}}\| + (\frac{1}{\gamma} + L)\|\Delta^k\| \cdot \|x^{k+1} - \overline{x^{k+1}}\|$$

$$+ \kappa \cdot \sum_{d=k+1-\tau}^{k} (d - (k-\tau))\|\Delta^d\|^2 + \langle \sum_{i=1}^{m} \nabla f_i(x^{k-\tau_{i,k}}) - v^k, x^{k+1} - \overline{x^{k+1}} \rangle + \lambda_k$$

$$\overset{a)}{\leq} \sum_{d=k-\tau}^{k-1} L\|\Delta^d\| \cdot \|x^{k+1} - \overline{x^{k+1}}\| + (\frac{1}{\gamma} + L)\|\Delta^k\| \cdot \|x^{k+1} - \overline{x^{k+1}}\| + \kappa\tau \cdot \sum_{d=k-\tau+1}^{k} \|\Delta^d\|^2$$

$$+ \|\sum_{i=1}^{m} \nabla f_i(x^{k-\tau_{i,k}}) - v^k\| \cdot \|x^{k+1} - \overline{x^{k+1}}\| + \lambda_k, \tag{52}$$

414 where $a)$ is from that $d - (k - \tau) \leq \tau$ when $k - \tau + 1 \leq d \leq k$. Let

$$a^k := \begin{pmatrix} \sqrt{\frac{1}{\gamma} + L}\|\Delta^k\| \\ \sqrt{L}\|\Delta^{k-1}\| \\ \vdots \\ \sqrt{L}\|\Delta^{k-\tau}\| \\ \sqrt{\kappa\tau}\|\Delta^k\| \\ \vdots \\ \sqrt{\kappa\tau}\|\Delta^{k-\tau+1}\| \\ \|\sum_{i=1}^{m} \nabla f_i(x^{k-\tau_{i,k}}) - v^k\| \\ \sqrt{\lambda_k} \end{pmatrix}, \quad b^k := \begin{pmatrix} \sqrt{\frac{1}{\gamma} + L}\|x^{k+1} - \overline{x^{k+1}}\| \\ \sqrt{L}\|x^{k+1} - \overline{x^{k+1}}\| \\ \vdots \\ \sqrt{L}\|x^{k+1} - \overline{x^{k+1}}\| \\ \sqrt{\kappa\tau}\|\Delta^k\| \\ \vdots \\ \sqrt{\kappa\tau}\|\Delta^{k-\tau+1}\| \\ \|x^{k+1} - \overline{x^{k+1}}\| \\ \sqrt{\lambda_k} \end{pmatrix}. \tag{53}$$

Using this and the definition of $F_k$ (8), we have:

$$(\mathbb{E}F_{k+1}(\varepsilon, t))^2 \leq \mathbb{E}[F_{k+1}(\varepsilon, t)^2] \leq \mathbb{E}\left|\langle a^k, b^k \rangle\right|^2 \leq \mathbb{E}\|a^k\|^2 \mathbb{E}\|b^k\|^2. \tag{54}$$

Direct calculation yields

$$\mathbb{E}\|a^k\|^2 \leq \max\{\frac{1}{\gamma} + L + \kappa\tau, 2D\} \cdot \left(\sum_{d=k-\tau}^{k} \mathbb{E}\|\Delta^d\|^2 + \phi_k^2\right) \tag{55}$$

and

$$\mathbb{E}\|b^k\|^2 \leq [(\tau+1)(\frac{1}{\gamma} + L) + 1]\mathbb{E}\|x^{k+1} - \overline{x^{k+1}}\|^2 + \kappa\tau \sum_{d=k-\tau}^{k-1} \mathbb{E}\|\Delta^d\|^2 + \lambda_k. \tag{56}$$

Thus, we prove the result.

**Proof of Theorem 4**

For a given $C > 0$, we set

$$\phi^k := \frac{C}{\sqrt{2\delta}}\zeta^k.$$

If $C$ is large enough, we have $\frac{\sigma_k}{\sqrt{2\delta}} \leq \phi_k$. Thus, for any $k$

$$\frac{\sum_{i=k}^{+\infty} \phi_i^2}{\phi_k^2} = \frac{1}{1 - \zeta^2}. \tag{57}$$

Therefore, Lemma 2 holds. It is easy to see that $\alpha(\kappa\tau \sum_{d=k-\tau}^{k-1} \mathbb{E}\|\Delta^d\|^2 + \beta\mathbb{E}\|x^{k+1} - \overline{x^{k+1}}\|^2 + \lambda_k)$ is bounded; and we assume the bound is $R$, i.e.,

$$\alpha(\kappa\tau \sum_{d=k-\tau}^{k-1} \mathbb{E}\|\Delta^d\|^2 + \beta\mathbb{E}\|x^{k+1} - \overline{x^{k+1}}\|^2 + \lambda_k) \leq R. \tag{58}$$

With Lemma 2, we then have

$$[\mathbb{E}F_{k+1}(\varepsilon, \delta)]^2 \leq R \cdot (\mathbb{E}F_k(\varepsilon, \delta) - \mathbb{E}F_{k+1}(\varepsilon, \delta)). \tag{59}$$

From [Lemma 3.8, [2]] and the fact $\mathbb{E}F(x^k) - \min F \leq \mathbb{E}F_k(\varepsilon, \delta)$, we then prove the result.

**Proof of Proposition 4**

In the deterministic case, Lemma 1 can hold without expectations, $\sup_k\{\xi_k(\varepsilon, t)\} < +\infty$, thus, $\sup_k\{F(x^k)\} < +\infty$ and $\sum_{d=k-\tau}^{k-1} \|\Delta^d\|^2 < +\infty$. Noting the coercivity of $F$, sequences $(x^k)_{k \geq 0}$ and $(\overline{x^k})_{k \geq 0}$ are bounded. Thus, $\left(\alpha(\kappa\tau \sum_{d=k-\tau}^{k-1} \|\Delta^d\|^2 + \beta\|x^{k+1} - \overline{x^{k+1}}\|^2 + \frac{\lambda_k^2}{\phi_k^2})\right)_{k \geq \tau}$ is bounded; and we assume the bound is $R$, i.e.,

$$\alpha(\kappa\tau \sum_{d=k-\tau}^{k-1} \|\Delta^d\|^2 + \beta\|x^{k+1} - \overline{x^{k+1}}\|^2 + \lambda_k) \leq R. \tag{60}$$

In the deterministic case, Lemma 2 can hold with deleting the expectation, we then have

$$F_{k+1}(\varepsilon, t)^2 \leq R(F_k(\varepsilon, t) - F_{k+1}(\varepsilon, t)). \tag{61}$$

From [Lemma 3.8, [2]] and the fact $F(x^k) - \min F \leq F_k(\varepsilon, t)$, we then prove the result.

**Proof of Theorem 5**

With (12), we have

$$\alpha\beta\mathbb{E}\|x^{k+1} - \overline{x^{k+1}}\|^2 \leq \frac{\alpha\beta}{\nu}(\mathbb{E}F(x^{k+1}) - \min F) \leq \frac{\alpha\beta}{\nu}F_{k+1}(\varepsilon, \delta) \leq \frac{\alpha\beta}{\nu}\mathbb{E}F_k(\varepsilon, \delta). \tag{62}$$

On the other hand, from the definition of (8),

$$\alpha\kappa\tau \sum_{d=k-\tau}^{k-1} \mathbb{E}\|\Delta^d\|^2 \leq \alpha\tau\mathbb{E}F_k(\varepsilon,\delta) \tag{63}$$

and

$$\alpha\lambda_k \leq \alpha\mathbb{E}F_k(\varepsilon,\delta). \tag{64}$$

Letting $H = \alpha\tau + \frac{\alpha\beta}{\nu} + \alpha$ and with Lemma 2,

$$[\mathbb{E}F_{k+1}(\varepsilon,\delta)]^2 \leq H(\mathbb{E}F_k(\varepsilon,\delta) - \mathbb{E}F_{k+1}(\varepsilon,\delta)) \cdot \mathbb{E}F_k(\varepsilon,\delta). \tag{65}$$

If $\mathbb{E}F_k(\varepsilon,\delta) = 0$, we have $0 = \mathbb{E}F_{k+1}(\varepsilon,\delta) = \mathbb{E}F_{k+2}(\varepsilon,\delta) = \ldots$. The result certainly holds. If $\mathbb{E}F_k(\varepsilon,\delta) \neq 0$,

$$\left(\frac{\mathbb{E}F_{k+1}(\varepsilon,\delta)}{\mathbb{E}F_k(\varepsilon,\delta)}\right)^2 + H\left(\frac{\mathbb{E}F_{k+1}(\varepsilon,\delta)}{\mathbb{E}F_k(\varepsilon,\delta)}\right) - H \leq 0. \tag{66}$$

With basic algebraic computation,

$$\frac{\mathbb{E}F_{k+1}(\varepsilon,\delta)}{\mathbb{E}F_k(\varepsilon,\delta)} \leq \frac{2H}{\sqrt{H^2 + 4H} + H}. \tag{67}$$

By defining $\omega = \frac{2H}{\sqrt{H^2+4H}+H}$, we then prove the result.

**Proof of Lemma 3**

1) If $\eta_k = \frac{2c}{(2\tau+1)L}$, we then have

$$F(x^{k+1}) - F(x^k) \leq \frac{L}{2\varepsilon} \sum_{d=k-\tau}^{k-1} \|\Delta^d\|^2 + \left[\frac{(\tau\varepsilon+1)L}{2} - \frac{(2\tau+1)L}{2c}\right]\|\Delta^k\|^2. \tag{68}$$

2) If the $\eta_k \geq \frac{2c}{(2\tau+1)L}$, with the Lipschitz continuity of $\nabla f$, we have

$$F(x^{k+1}) - F(x^k) \leq \langle\nabla f(x^k), x^{k+1} - x^k\rangle + g(x^{k+1}) - g(x^k)$$

$$= \underbrace{\langle\sum_{i=1}^{m}\nabla f_i(x^{k-\tau_{i,k}}), x^{k+1} - x^k\rangle + g(x^{k+1}) - g(x^k)}_{\dagger}$$

$$+ \langle\nabla f(x^k) - \sum_{i=1}^{m}\nabla f_i(x^{k-\tau_{i,k}}), \Delta^k\rangle + \frac{L}{2}\|\Delta^k\|^2. \tag{69}$$

Based on the line search rule, $\dagger \leq -\frac{c_2}{2}\|x^{k+1} - x^k\|^2$, thus,

$$F(x^{k+1}) - F(x^k) \leq \langle\nabla f(x^k) - \sum_{i=1}^{m}\nabla f_i(x^{k-\tau_{i,k}}), \Delta^k\rangle + \frac{L - c_2}{2}\|\Delta^k\|^2$$

$$\overset{a)}{\leq} \|\nabla f(x^k) - \sum_{i=1}^{m}\nabla f_i(x^{k-\tau_{i,k}})\| \cdot \|\Delta^k\| + \frac{L - c_2}{2}\|\Delta^k\|^2$$

$$\overset{b)}{\leq} (L\sum_{d=k-\tau}^{k-1}\|\Delta^d\|) \cdot (\|\Delta^k\|) + \frac{L - c_2}{2}\|\Delta^k\|^2$$

$$\overset{c)}{\leq} \frac{(\tau\varepsilon+1)L}{2}\|\Delta^k\|^2 + \frac{L}{2\varepsilon}\sum_{d=k-\tau}^{k-1}\|\Delta^d\|^2 - \frac{c_2}{2}\|\Delta^k\|^2, \tag{70}$$

where $a)$ is from the Cauchy inequality, $b)$ is the triangle inequality, $c)$ uses the Schwarz inequality.

By setting

$$\frac{1}{\varepsilon} + \varepsilon = 1 + \frac{1}{\tau}\left(\frac{2\tau+1}{2c} - \frac{1}{2}\right),$$

with direct computation, for both case,

$$\xi_k(\varepsilon) - \xi_{k+1}(\varepsilon) \geq \frac{1-c}{4c}(L + 2\tau L) \cdot \|\Delta^k\|^2. \tag{71}$$

That means

$$\lim_k \|\Delta^k\| = 0. \tag{72}$$

Noting that (28) still holds here. With (72) and (28), we then derive the result.

## Footnotes

[1]We say a sequence $a^k$ is in $\ell^1$ if $\sum_{k=1}^\infty |a^k| < \infty$.