[Reviews · NeurIPS 2019]

Reviewer 1



Originality The PIAG algorithm has attracted a lot of attentions in recent years, but few of them use Lyapunov function for analyzing. As a result, I think the originality is good enough for accept. Quality The main contribution lies on the analysis. The proof is neat, but there are some typos, especially in taking expectation. I think the authors should spend a few words to clarify which r.v. is taking expectation. The experiment is weak. There is no baselines for comparison. Clarity The clarity is also hurt by the issue in taking expectation. I think the authors could fix them in the revision. The authors should spend more words on KL property. There are quite a lot of references on KL property, e.g.[1][2]. But I cannot find any of them. Significance i. Convergence rate. The non-asymptotic rate achieve by assuming 1) boundedness of $\{x_t\}$; 2) semi-algebraic of objective, which is a sufficient condition for KL property. This is not new in optimization community. ii. Lyapunov function. I think the main contribution is that this is the first paper for analyzing PIAG by Lyapunov function. iii. Experiment. The experiment is not convincing. The authors should include more comparisons in the revision, e.g. [3]. iv. The paper is mainly based on the assumption that $\sum_i \sigma^2_i<\infty$. However, the authors did not provide a method to guarantee the assumption in stochastic setting. This dramatically hurt the contribution of the paper. Overall, I would upgrade my score if the authors could give a feasible plan for fixing above issues. [1] Bolte, Jérôme, Shoham Sabach, and Marc Teboulle. "Proximal alternating linearized minimization for nonconvex and nonsmooth problems." Mathematical Programming 146.1-2 (2014): 459-494. [2] Attouch, Hédy, et al. "Proximal alternating minimization and projection methods for nonconvex problems: An approach based on the Kurdyka-Łojasiewicz inequality." Mathematics of Operations Research 35.2 (2010): 438-457. [3] Peng, Wei, Hui Zhang, and Xiaoya Zhang. "Nonconvex Proximal Incremental Aggregated Gradient Method with Linear Convergence." Journal of Optimization Theory and Applications (2018): 1-16. __________________________________ I have read the authors' feedback. I believe most of the issues will be fixed in the revision. The error summable assumption is still a challenge. I think we can make it hold in certain cases, but more analysis might be required. I think the paper is significant enough for the conference. Thus I’d like to upgrade my score.

Reviewer 2



The submission provides an interesting framework coupled with Lyapunov-based methods to build a unified framework for IAG methods. The idea is interesting and appealing. However, the submission suffers from a lack of clarity that is damaging these ideas: - many definitions are lacking (e.g., the semi-algebraic property and the distance used at l. 154 should be at least commented); - results are commented but I feel some assumptions should be discussed (e.g., l. 192, is this assumption realistic when \sigma_k = k^{-\eta}?); - plots are hard to read and would benefit from additional comments. Overall, the submission seems like it brings valuable ideas but needs additional work around the results and assumptions discussions (the related work sections are nicely written to that regard). Updated review: It appears that I did not understand well the novelty of some points of the paper. I intend to increase my score, but I still think the paper severely lacks clean statement and discussions about the results and their comparison to existing work.

Reviewer 3



Originality: This paper presents a general PIAG algorithm an inexact extension of previously proposed exact PIAG from [1]. The paper analyzes convergence properties of this algorithm under different assumptions of convexity of the component functions. This is an extension of [1], [2] where linear convergence was proven under strong convexity. The paper also adds a dimension of line search and proves larger stepsizes than earlier. The Lyapunov function based proof technique has been used earlier in [1] and [2], however, the perspective of convergence in expectation, almost surely, and semi-algebraic property is a new addition. Quality: The paper contains high quality results and proofs. Although, the primary focus of the paper is on theory, the quality of numerical section and the figure can be much improved. Clarity: The paper is overall well written and the contributions and organization are explained clearly upfront. However, there are few places where concepts or quantities are used without defining them clearly. For example, * Line 132: can be explained better or add citation to the result if taken from somewhere else * Line 150: the assumption on sigma_i’s can be explained better with some discussion/note (since inexactness is the main difference in the algorithm between this paper and [1]) * Line 191, Lemma 2: uses a quantity ‘t’ without defining it * Line 210: uses “coercivity of f” without defining it * Line 439, Theorem 5: uses incorrect solution of quadratic equation, the inequality should be LHS <= 0.5*( \sqrt(H^2+4H) - H) Significance: This paper extends the existing work on convergence of PIAG and related methods to non-strongly convex and nonconvex areas. The extension with line search makes this algorithm more applicable to real problems when convergence times (vs. theoretical rates) matter more. [1] https://arxiv.org/pdf/1608.01713.pdf [2] https://www.di.ens.fr/~fbach/Defazio_NIPS2014.pdf

[Author Response · NeurIPS 2019]

**1 Rebuttals for "General Proximal Incremental Aggregated Gradient Algorithms: Better and Novel Results un-**
**2 der General Scheme"**

We thank the three reviewers for their constructive feedback.

**4 Reviewer 1.**

**Q1**:*... a lot of references on KL property, e.g.[1][2]. But I cannot find any of them.* **A1**: Thanks for your kind
suggestions. [1][2] are good references for the KL property and should be included in the further version of our paper.
The detailed definitions and references of KL and semi-algebraicity will be presented in the appendix part.

**Q2**: *The non-asymptotic rate achieve by assuming 1) boundedness of $\{x_t\}$; 2) semi-algebraic of objective, which is*
*a sufficient condition for KL.* **A2**: Good suggestion. As R1 says, the non-asymptotic rate can be proved under these
conditions. The proofs can be presented with existing methodology given in works like [1][2]. The future version can
present the results about the rates.

**Q3**:*The experiment is not convincing. The authors should include more comparisons in the revision, e.g. [3].* **A3**: We
will add the numerical comparisons with other incremental gradient methods including the one given in [3] for convex
and nonconvex regression tasks. Thank you!

**Q4**:*The paper is mainly based on the assumption that $\sum_i \sigma_i^2 < \infty$. However, the authors did not provide a method to*
*guarantee the assumption in stochastic setting.* **A4**: The assumption $\sum_i \sigma_i^2 < \infty$ is like the Lipschitz assumption of
gradient. If a function is nonsmooth, the vanilla gradient descent certainly cannot work. The theory built in this paper is
for the algorithms satisfying the summable assumption. But if the algorithm fails to obey, the general PIAG cannot
work, either. Whether the assumption is satisfied or not depends on the algorithm itself. In lines 131-136, we mentioned
these facts. This is no universal way to guarantee the assumption just like that we cannot make sure all functions are
smooth. For example, for SGD, SVRG and SAGA, the assumption is broken. But for the stochastic BCD algorithm and
the asynchronous SBCD, the assumption holds well. We can provide the guarantee for block coordinate descents. And
we will be specific on this point in future version.

**24 Reviewer 2.**

**Q1**: *many definitions are lacking (e.g., the semi-algebraic property and the distance used at l. 154)* **A1**: We will give
the detailed definitions of KL and semi-algebraic in the appendix. The distance is denoted by $\text{dist}(\mathbf{0}, \partial F(x^k)) :=$
$\min_{v \in \partial F(x^k)} \|v\|_2$. We will be specific on the definitions and notation in revision. Thanks.

**Q2**: *I feel some assumptions should be discussed (e.g., l. 192, is this assumption realistic when $\sigma_k = k^{-\eta}$?)* **A2**:
$\sigma_k = k^{-\eta}$ cannot obey the assumption. The assumption can promise geometric decreasing parameters, which are given
in Theorem 4 and 5. We currently cannot include the assumption $\sigma_k = k^{-\eta}$ for technical reasons. How to weaken the
assumption will be left to future studies.

**Q3**: *plots are hard to read and would benefit from additional comments.* **A3**: The figure will be enlarged and additional
comments (like comparison and explanations of the performances) will be given. We will add experiments with other
incremental methods as mentioned in A2 for Reviewer 1.

**35 Reviewer 3.**

**Q1**: *The paper contains high quality results and proofs. Although, the primary focus of the paper is on theory, the*
*quality of numerical section and the figure can be much improved.* **A1**: Thank you very much for your positive opinions.
The extra numerical experiments and additional comments will be made and plots will be enlarged. Please see A3 to R1
and A3 to R2.

**Q2**: *Line 132: can be explained better or add citation to the result if taken from somewhere else.* **A2**: The result can be
found in [Neterov Y. 2011, Efficiency of coordinate descent methods on huge-scale optimization problems, Siam J.
Opt.] and [25]. We will add the citations.

**Q3**: *the assumption on $\sigma_i$'s can be explained better with some discussion/note.* **A3**: Thank you for your suggestion.
We answered part of this in A4 for R1. And we will be more specific on this assumption.

**Q4**: *Lemma 2: uses a quantity 't' without defining it.* **A4**: That is a typo. Actually, here $t = 1$. We will correct it.

**Q5**: *(the inequality should be LHS $\leq 0.5 \times \sqrt{H^2 + 4H} - H$* **A5**: We do not compute errors here because $\frac{\sqrt{H^2+4H}-H}{2} =$
$\frac{(\sqrt{H^2+4H}-H)(\sqrt{H^2+4H}+H)}{2(\sqrt{H^2+4H}+H)} = \frac{4H}{2(\sqrt{H^2+4H}+H)} = \frac{2H}{(\sqrt{H^2+4H}+H)}.$

**All reviewers:** We will address your other comments in the final version. All of your major concerns have been
addressed above. We hope you can reconsider your opinion on our paper.

[Meta-Review · NeurIPS 2019]

Although there was no consensus reached, overall there were more positive points than negative points brought up in the reviews. I am recommending a weak acceptance, please take into account the reviewer comments in the next revision as they will improve the quality of the paper.